# Using knowledge translation to support the use of evaluation findings: A case study of the linda mama free maternity program in Kenya

**Fatuma H. Guleid**[1]*, **Stacey Orangi**[1], **Angela Kairu**[1], **Brian Arwa**[1], **Janet Keru**[2],
**Anne Musuva**[2], **Ileana Vilcu**[3], **Anooj Pattnaik**[4], **Nirmala Ravishankar**[5],
**Edwine Barasa**[1,6]

**1** Health Economics Research Unit (HERU), KEMRI-Wellcome Trust Research Programme, Nairobi, Kenya,
**2** ThinkWell, Nairobi, Kenya, **3** Thinkwell, Geneva, Switzerland, **4** ThinkWell, Washington DC, United States
of America, **5** Bill and Melinda Gates Foundation, Washington DC, United States of America, **6** Nuffield
Department of Medicine, Centre for Tropical Medicine and Global Health, University of Oxford, Oxford, United
Kingdom

* fguleid@kemri-wellcome.org

journal.pgph.0003961

Tropical Medicine: Instituut voor Tropische
Geneeskunde, BELGIUM

**Data Availability Statement:** Public deposition of
the transcripts would breach compliance to the

## Abstract

Using program evaluation findings is crucial in improving health programs and realising the
program's benefits. In this article, we report on how a knowledge translation (KT) approach
supported the use of evaluation findings to improve the Linda Mama free maternity program
in Kenya. We used a case study design employing qualitative approaches to describe our
KT strategy and its impact on evaluation use. Data were collected through semi-structured
in-depth interviews of participants (n = 25) in three Kenyan counties following dissemination
of the evaluation findings and co-production of action plans based on the evaluation. The
findings suggest modest improvements in the implementation of Linda Mama in 3 Kenyan
counties facilitated by application of the evaluation findings. However, these improvements
were not uniform across and within the counties. Challenges such as the COVID-19 restric-
tions, lack of infrastructure and delayed reimbursement of funds hindered the full implemen-
tation of the action plans. The KT strategy was a key facilitator for the improvements. The
dissemination and deliberation workshops provided learning spaces for stakeholders,
ensuring that each perspective was considered. The participatory method used in develop-
ing the action plans also improved communication between stakeholder groups. Partici-
pants reported that this approach made aware them of the gaps in implementation and
motivated them to realise the full potential of the Linda Mama program. Using KT, especially
when evaluating and refining the implementation of complex health programs with multiple
stakeholders, is useful in improving the uptake of evaluation findings. However, it can be
challenging to sustain such engagement with stakeholders. In addition, contextual factors
that affect uptake need to be considered and navigated. Finally, significant investment (both
in human resource and financial) in such approaches is required if KT is to be successful.

approved study ethics protocol. De-identified excerpts of the transcripts relevant to the study can be available upon reasonable request from the KEMRI-Wellcome Trust Research Programme's data governance committee through the email: dgc@kemri-wellcome.org.

**Funding:** This work was funded by the Strategic Purchasing for Primary Health Care (SP4PHC) project, which is supported by the Bill & Melinda Gates Foundation and implemented by ThinkWell (BM, JK, IV, AM, AP, NR) in collaboration with learning partners including KEMRI Wellcome Trust. Additional funds from a Wellcome Trust core grant (EB, AK, SO, BA, FHG) awarded to the KEMRI-Wellcome Trust Research Program (#092654) supported this work. The funders had no role in study design, data collection and analysis, decision to publish, or preparation of the manuscript.

**Competing interests:** The authors have declared that no competing interests exist.

## Introduction

Considering the resource demands of implementing wide-scale health programs, there is particular impetus to ensure that these programs provide the anticipated benefits to society. Health program evaluation provides information on whether a program is working, why and under what circumstances. In addition, evaluation findings help provide knowledge that improves the effectiveness and efficiency of existing health programs. However, this knowledge is only useful if it is utilised by stakeholders (intended users of the evaluation) to improve the health program.

The field of evaluation has long focused on the use of evaluation findings as an indicator of successful evaluation [1–3]. Several factors have been reported to affect the use of evaluation findings [4, 5]. In particular, including stakeholders in the evaluation through participatory evaluation (PE) is believed to strengthen the use of evaluation findings through several mechanisms [6–9]. For example, incorporating the needs of stakeholders enables them to develop their own sustainable and context-specific solutions to the issues they face during program implementation [8]. In addition, in multi-stakeholder programs, PE increases collaboration between different stakeholder groups, highlighting their diverse views and promoting a greater understanding between them. Furthermore, PE increases the capacity for stakeholders to address the challenges and gaps within the program implementation by being a part of the learning and assessment process [10, 11]—however, problems with utilising evaluations persist [4]. In addition, there is little evidence on which strategies support the uptake of evaluation findings in complex decision-making contexts such as health systems.

The field of knowledge translation (KT) can provide insights on improving evaluation uptake [12]. KT can be defined as the "methods for closing the gaps from knowledge to practice" [13]. Its overall purpose is to promote research uptake during clinical and policy decision-making to provide effective health services and polices for improved health outcomes [14, 15]. However, while "knowledge" is often implied to mean research evidence, it can also mean knowledge derived from other sources, including evaluations. Although the literature on both KT and evaluation use has grown in parallel, they describe similar change processes [16]. Some effort has gone into bringing these fields together. For example, Amo and Cousins explored how KT-related theories can support evaluation use and highlighted the importance of using developments from the knowledge use field to inform the evaluation field [17]. Furthermore, Donnelly and Searle suggest that KT can support evaluation use in three ways: (i) by synthesising surrounding information for robust evaluation findings, (ii) by ensuring that findings are packaged in an accessible and useable manner and (iii) that the evaluation is conducted with the users in mind [12]. In a case study assessing the role KT plays in the use of evaluations in foundations and non-governmental organisation, KT was reported to enhance use of evaluations by promoting understanding, facilitating decision-making, and improving program effectiveness [18]. Thus, KT is a natural extension of the interest in use of evaluations and can lead to an increased understanding of how evaluation findings can be used to improve health programs.

Several KT models and approaches have been described, including push/pull models and more dynamic and interactive approaches such as integrated knowledge translation [19]. Integrated knowledge translation (iKT) is an approach to knowledge translation that focuses on the engagement of knowledge users throughout the research process [20]. Successful iKT is believed to improve research use in policy and practice settings based on the idea that difficulties with research uptake are not just about the *transfer* of research findings but rather about *how* these findings are produced [20, 21]. An iKT strategy promotes collaborative inquiry between researchers and knowledge users, which aims to produce research that addresses the

priority issues of knowledge users [21]. A shared perspective in the iKT and PE approaches is that research and evaluation findings are more likely to be used if users are actively involved in the research or evaluation process. The literature on how KT can support the uptake of evaluation findings for improved health programs is still growing. In this paper, we sought to contribute to this body of evidence by describing an iKT strategy for increasing the uptake of the findings from a process evaluation of a health program in Kenya and its impact. While the design of the evaluation and its outcomes are discussed to provide context, the primary focus of this paper remains on how the KT strategy facilitated the use of evaluation findings.

## Context of the case

A process evaluation of the Linda Mama free maternity program in Kenya provides the context in which we describe our iKT-informed evaluation.

### The Linda Mama free maternity program

In Kenya, poor utilisation of skilled maternal care was found to be a key contributor to the high rate of maternal mortality [22]. This has been attributed partly to the unaffordability of maternal health services. To address this, the Kenyan Government introduced a policy providing free maternity care, removing user fees for maternal healthcare in all public healthcare facilities in 2013 [23]. An initial evaluation of the implementation of the policy highlighted several challenges [24]. For example, some services such as antenatal and postnatal care were not provided due to the lack of clarity on the benefits package. In addition, reimbursement of funds to facilities was also routinely delayed, which meant that facilities could not sustainably provide services. To address these issues and improve accountability and efficiency, the Government revised the program in 2016. It moved its management from the Ministry of Health (MoH) to the National Hospital Insurance Fund (NHIF) [25]. The revised policy, dubbed "Linda Mama", extended access to include faith-based and private providers to reduce the strain on public facilities. It also expanded the benefits package to include antenatal and postnatal care explicitly. However, at the time of this study, there were no empirical studies on whether/how these revisions have improved the implementation of the free maternity policy. To this effect, we carried out a process evaluation of the Linda mama program. The findings of the process evaluation have been published separately [26]. In summary, the process evaluation found that while Linda Mama resulted in improved accountability and expanded benefits, beneficiaries did not access some services that were part of the revised benefits package. Beneficiaries were still incurring out-of-pocket payments, and because of the devolution of governance from the national Government to the county governments, health facilities in most counties had lost financial autonomy and had no access to Linda Mama reimbursements from NHIF for services provided. In addition, fund disbursements from NHIF were characterised by delays and unpredictability. Finally, there was inadequate communication between stakeholders, challenges with processing Linda Mama claims at facilities and reimbursement rates were deemed insufficient.

## Methods

### Integrated knowledge translation approach

Our iKT strategy for improving the implementation of the Linda Mama program in three Kenyan counties embedded KT principles at every stage of the evaluation and involved three steps (Fig 1). Our strategy was informed by Smits and Champagne's model on practical participatory evaluation which emphasises active stakeholder involvement throughout the evaluation

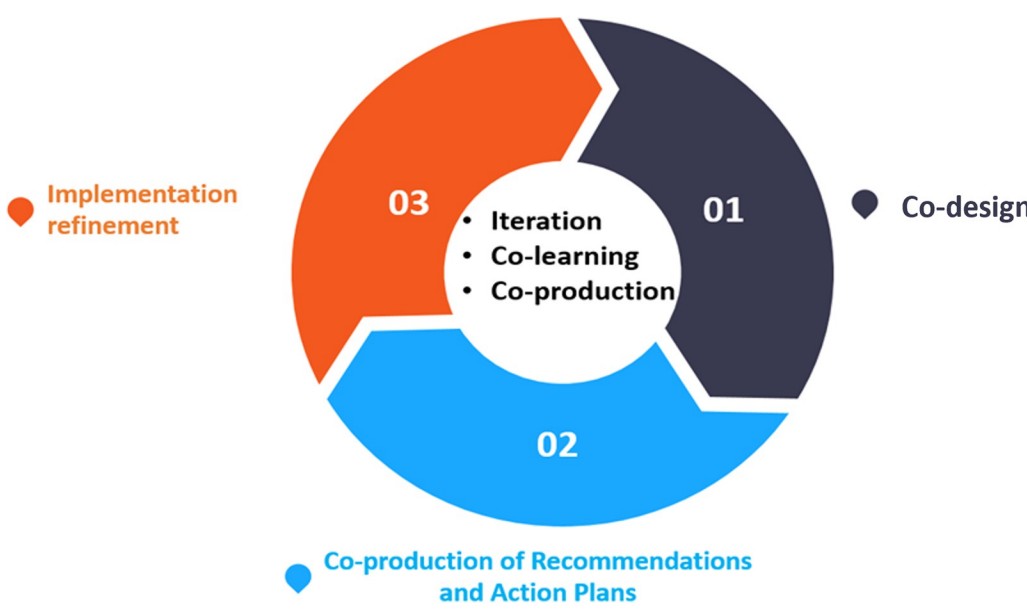

**Fig 1. The study's iKT approach.**

process [9]. The model posits that four key concepts underpin evaluation use in PE. These include interactive data production, knowledge co-construction, local context of action and instrumental use. Based on the model, the likelihood of evaluation use increases because stakeholders feel a sense of ownership over the evaluation.

**Step 1. *Co-design*.** The research team facilitated two planning meetings involving researchers, implementers and other stakeholders within the National Linda mama technical working group at the evaluation design stage. The stakeholders were asked to identify the key evaluation objectives and questions during these meetings. They were also asked to discuss broad evaluation design issues such as sampling selection. This process of co-design of evaluation questions ensured that the stakeholders' unique experiences and evidence needs were considered to inform the design and conduct of the process evaluation.

**Step 2. *Co-production of recommendations and action plans*.** To facilitate knowledge co-construction, we presented the process evaluation findings to all the relevant stakeholders once the evaluation was complete. The researchers organised a one-day workshop in February 2020 for each participating county (3 workshops in total). The objectives of these workshops were to exchange the findings of the process evaluation, ask stakeholders to discuss and validate the findings, and, based on the discussions, develop action plans to address the challenges identified during the evaluation. A total of 105 stakeholders attended the workshops.

The workshops employed a participatory approach and were structured around three activities:

1. **Presentation of evaluation findings:** Researchers presented the findings from the process evaluation to the stakeholders followed by a question-and-answer session to provide clarification.

2. **Group session 1:** *validation of findings*: stakeholders were put in role-specific subgroups (e.g., all health providers formed one subgroup). The subgroups were then asked to answer two questions: (a) what refinements of the findings would they recommend? and (b) which challenges with Linda Mama have not been captured by the process evaluation?

3. **Group session 2:** *develop action plans*: the subgroups were asked to select 3–4 priority challenges identified from the evaluation. For each of the priority challenges, the groups were asked to:

   a. Identify a feasible, actionable solution to each challenge

   b. Develop a timeline for its implementation

   c. Identify an indicator for monitoring its implementation

   d. Identify what resources will be needed to implement the solution

   e. Identify who will be responsible for implementing the solution

**Step 3. *Implementation refinement.*** In this step, the implementing stakeholders were expected to implement their action plans to improve the implementation of the Linda Mama program in their counties. In all three counties, technical assistance was offered to the counties by a development partner (ThinkWell). We measured the success of the iKT strategy by tracking the implementation of the action plans. To this effect, we conducted in-depth interviews to assess the extent to which these action plans had been implemented and the implementing stakeholders' experiences after 19 months. Initially, the implementation period was set for one year. However, data collection was delayed due to COVID-19 restrictions. We also collected data on the utility of the iKT approach and its impact on improving Linda Mama implementation. The project timeline is shown in Fig 2.

## Study sites (geographical)

The study was conducted in 3 purposively selected Kenyan counties. These counties were chosen to represent different regions of the country and have different proportions of skilled birth deliveries and shares of urban-rural population. The selected counties included two sites with universal health coverage (UHC) initiatives and which stakeholders were keen to understand Linda mama implementation in parallel to other UHC initiatives. The third county was purposively selected because previous work on the program has been done in this county, and links to the county government and some health facilities had been established.

## Study population

The study population was drawn from the national, county, and health facility levels who had the responsibility to implement the action plans (implementing stakeholders). At the national

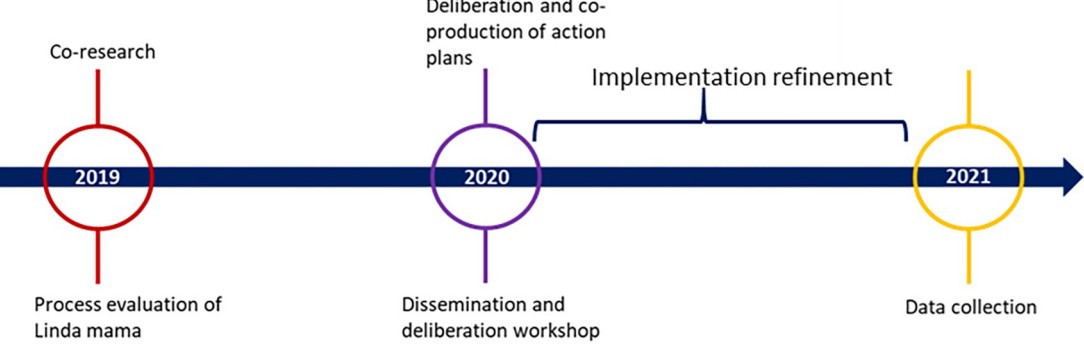

**Fig 2. Timeline of the project.**

**Table 1. Participants included in the study.**

|  | COUNTY A | COUNTY B | COUNTY C |
|---|---|---|---|
| COUNTY OFFICIALS | 2 | 2 | 2 |
| NHIF OFFICIALS | 2 | 1 | 1 |
| HEALTH PROVIDERS | 4 | 6 | 5 |

level, the study population comprised key informants from the NHIF- officials at the national headquarters and county branches. At a county level, the study respondents were officials from the health departments of the selected three counties who were familiar with Linda Mama's implementation process, managed funds and monitored mechanisms of the Linda Mama program. Within the facilities, participants included the facility-in-charge at facilities offering maternity services. Only implementing stakeholders who attended the co-research meetings and workshops were selected for data collection (purposive sampling) (Table 1).

## Data collection

Data were collected through semi-structured, in-depth interviews. The interview questions assessed the following issues: (a) progress on the implementation of the co-developed action plans, (b) outcomes of implementing the action plans and (c) usefulness of the iKT approach in supporting improved implementation of Linda Mama. Interviews were conducted by four researchers (SO, BA, AK and FHG). Each interview lasted 30–60 minutes and was audio-recorded using an encrypted audio recorder. Participant recruitment and data collection were done concurrently from October 4th–October 29th 2021.

## Data management and analysis

All audio recordings were transcribed verbatim in MS Word. The transcripts were read and reread (by FHG and SO) to familiarise ourselves with the data. We began our analysis by conducting open coding where key concepts and patterns were identified through and inductive process. Subsequent rounds of coding were more focused and based on the three key issues of investigation highlighted in the interview guide. These key issues served as our primary themes. Any emerging sub-themes from the primary themes were noted. After these initial rounds of coding, we developed a coding framework which was iteratively refined as coding progressed. All predefined themes and any other additional themes were captured within the framework. To ensure rigour during the coding process, coding was conducted by 2 coders (FHG and SO) independently. Both coders met regularly to discuss the coding framework and codes. All transcripts were coded using NVIVO qualitative data analysis software (QSR International Pty Ltd. Version 12, 2014).

## Ethical considerations

This study received ethical approval from the Kenya medical research institute (KEMRI) Scientific and Ethics Review Unit. Information sheets detailing the study description were shared with all participants before the interviews began after which signed informed consent was obtained.

## Results

Overall, participants noted that the iKT strategy contributed to the use of the evaluation findings through the implementation of the evaluation-informed action plans. The inclusion of

stakeholders from the outset of the evaluation process ensured that the evaluation was aligned to their needs and priorities which in turn facilitated the co-development of contextually appropriate action plans. In addition, the strategy provided a platform for exchanging findings and allowed stakeholders to engage directly with each other and share their perspectives. Participants reported that embedding the KT strategy within the evaluation, informed them of the gaps in implementation and motivated them to realise the full potential of the Linda Mama program. In this results section, we present the impact of the iKT strategy, the key aspects that worked and the challenges we faced.

### Impact of the iKT strategy

To measure the impact of our iKT strategy, we assessed the extent to which the evaluation-informed action plans had been implemented and the impact this had on Linda Mama. S1 Table outlines the co-created action plans and the progress on their implementation at the time of data collection in more detail.

Across the counties, the implementation of action plans varied suggesting that some of the findings of the evaluation were utilised directly to improve the Linda Mama program. For the actions that were implemented, the participants highlighted several outcomes. For example, one of the actions in the action plan was to increase sensitisation and training by NHIF on Linda mama in health facilities. Implementation of this led to an increased number of facilities claiming Linda Mama benefits across the three counties. Consequently, the number of reimbursements from NHIF to the facilities increased.

*".at least now we are receiving some [Linda mama] claims. I can tell you we've almost doubled the collections,"* KWTR0050 NHIF branch manager, County C.

*"The reimbursements from NHIF have grown by almost 30%."* Hospital accountant, County B

*"So, out of the 11 facilities we visited, we've seen [Linda Mama] claims from eight, and we are happy."* NHIF Branch Manager, County A

Additionally, the number of mothers registering for and utilising Linda Mama services increased as outreach programs and health talks increased their awareness of the program.

*"Currently, you know with the sensitisation now we are able to bring many people on board and when they get to know about the benefits of the Linda Mama package, they always come to us for registration."* NHIF branch manager, County C

*". . .it has really increased the numbers. Because before you found most of the mothers were not aware of Linda Mama. A mother could come to you and ask you what is Linda Mama? What about it? But for now, because we are giving them health talks, we are using the community health volunteers, and everyone is reaching out to these mothers, the numbers have increased".* Nurse-in-charge, County C

For County A, there was improved engagement and collaboration between the county officials and NHIF, which was a key facilitator in supporting implementation.

*"NHIF has been very supportive, I think work has just been very easy, especially for the public facilities."* Deputy County Director of Health, County A

## Key aspects of the iKT strategy that worked

**Workshops as a dissemination &deliberation tool.** The workshops provided physical spaces for the findings of the evaluation to be shared and discussed with and amongst stakeholders. By participating in the workshop, they could share their perspectives on the implementation of Linda Mama and reach a mutual understanding on the gaps in implementation and the actions needed to address these gaps:

*"It was helpful in the sense that it brought all the stakeholders to the sub-counties together. It's also useful that it helped us to identify those challenges. And it is from the workshop we got the insight that we can claim for this [Linda mama] money"* Referral hospital Matron, County A

*"At that particular time the workshop actually reached its objectives. We were able to pinpoint where the challenges and the gaps were from our side and I think NHIF were also able to point out the challenges from their side"* Chief Officer, County Department of Health, County A

In addition, the workshop facilitated communication between the stakeholders and encouraged engagement between them, something they were not doing adequately before with regards to the implementation of Linda Mama:

*"It's actually after that [the workshop], that we've had multiple meetings with NHIF on Linda Mama. It was all triggered by that workshop. That was the trigger; before that, there was nothing much, we just wrote letters and sat back."* Deputy Director, County Department of Health, County A

*"After the workshop, we have seen some improvement in the way we are doing it. Also, it was kind of a push because I think we couldn't think of engaging the Ministry of Health because of the limitations in benefits. So, it really opened a platform for us to go and engage with the ministry and with a reason for that because at least we had a place where we are coming from."* NHIF Branch manager, County C

Finally, as a dissemination activity, the workshop was considered a better alternative to passive dissemination of evaluation findings such as dissemination of written documents or pamphlets. Active dissemination, which encouraged in-person engagement, allowed stakeholders to gain a better understanding of the evaluation findings.

*"The way the workshop was structured in terms of information sharing, with the presentations and the like. . . it enabled us to understand what was inside as opposed to if you had sent us what. . .pamphlets, policy documents and that, but this was an engaging workshops"* Chief Officer, County Department of Health, County B

**Feedback mechanism for identifying gaps in implementation.** Participants also felt that the feedback from the process evaluation provided during the workshops was useful in improving implementation in two ways. First, it highlighted the current gaps with implementation, which allowed stakeholders to prioritise and act on by developing action plans.

*"Of course, because, it's through the feedback that we were able to build and develop the action plans."* Director, Health planning, County C

Second, it provided the stakeholders with an evidence-informed baseline to track their progress and measure their performance.

*"That external input is very important. One is you get to engage yourself as a manager as well, am I doing the right things? Am I doing what is expected of me? Maybe am I doing enough? Actually, the biggest challenge is, are you doing enough? Or is there more that I can do or is expected to me? So, all that is good input for a manager. It helps us know where we are and what we need to do."* Deputy Director, County Department of Health, County A

*"Having that research component for a project implementation is key because sometimes you do things, just assuming you are doing the right things and all that but only research gives you facts."* Chief Officer, County Department of Health, County B

## Challenges and limitations to the iKT strategy

One of the key challenges we faced in implementing this iKT strategy was our inability to sustain a high level of engagement throughout the implementation period. This was highlighted by participants who felt that they would prefer regular and frequent engagement with researchers who could provide more support in addressing challenges in real-time:

*"I think we should have had another one maybe after two quarters of this year so that we see ourselves if there is any improvement from where we were? Is there something that we are doing right?"* Medical superintendent, County B

*"The researchers need to not to exit the scene, because they have their insights of the issues. They need to assist us in addressing the issues in a way that is sustainable."* Director, Health Planning, County C

The other key challenge were the contextual factors that hindered the full implementation of the action plans. The emergence of COVID-19 and the subsequent restrictions meant that activities such as forums and meetings promoting engagement and building awareness of Linda Mama could not be done. In addition, the number of patients seeking services decreased due to the stigma around COVID-19.

*"One thing that has really affected us not to fully implement what we had chosen as our resolutions, you know after Covid-19 came, there was the issue of social distancing and all that sort of confusion, and we really lost many of our clients during that period and there are some programs that really needed close interactions and because of this issue of social distancing it became an issue to implement"* Medical Superintendent, County B.

The lack of sufficient infrastructure to support implementation posed more challenges. In facilities in one of the counties, there were issues with electricity, internet, lack of equipment such as computers and printers, and human resources to submit Linda Mama claims. NHIF and providers in all three counties added that the high staff turnover due to short-term contracts meant that the capacity to submit Linda Mama claims was not retained, and facilities would need to train new staff frequently.

*"There are a number of challenges that have come up in terms of space, human resource, staff are claiming they're very busy, attending to patients this is an additional task that is wearing them down. The required computers and, communication infrastructure, the copiers, all that*

*are not in place. And the biggest challenge right now is that for this financial year, we could not get any funding to support that installation"* Deputy County Director of Health, County A

"*The other thing is about the re-sensitisation, this is majorly affected by the high turnover of the casuals in these facilities, because you'll find that once we train them, within the next three months, another group comes on board."* NHIF branch manager, County C

In addition, delayed reimbursement from NHIF to facilities was a persistent challenge across all three counties.

"*Between November last year and July, around June, July, there were no, actually there were no reimbursements, claims were going but there were no reimbursements."* County Department of Health officer, County B

These contextual challenges do not reflect the failure of the iKT strategy itself but underscores the complexity of translating evaluation findings into practice in dynamic and unpredictable contexts.

## Discussion

It is becoming increasingly evident that approaches where researchers and knowledge users form partnerships have more impact on the uptake of research evidence [27, 28]. In our study, researchers engaged with stakeholders throughout the evaluation process from the design of a process evaluation, including setting objectives and the overall design (sample, population, etc.) to the co-development of evaluation informed action plans. The goal was to create an environment where stakeholders could actively participate in the design, interpretation and application of evaluation findings. This ensured that the evaluation objectives aligned with stakeholders' views and needs which has been shown to facilitate evaluation use in some studies [29]. In addition, the workshops acted as learning and sharing spaces in which knowledge (evaluation findings) was co-constructed and transformed into actionable plans. Participants agreed that the workshops increased participation and encouraged multidirectional knowledge exchange between them and researchers and amongst themselves. Similarly, increased stakeholder involvement which promoted communication was reported to lead to greater use of evaluation findings [30–32]. Development of actionable, evidence-informed recommendations not only demonstrated instrumental use of the evaluation findings but has also been reported to increase use [33, 34].

Contextual factors significantly influenced the uptake of findings independent of the iKT approach. For instance, the COVID-19 control measures shifted policy and funding priorities and hindered stakeholder engagement. In addition, the lack of parallel improvements at the systems level (e.g. availability of funds, availability of infrastructure, human resource issues, etc.) meant that some solutions could not be implemented at the facility level. This points to the importance of applying a health systems lens to implementing programs such as Linda Mama if they are to be successful [35, 36]. In addition, deeper engagements with top-level policy makers who control funding and infrastructure decisions might have mitigated this as suggested by Barrios (1986) who reported that recommendations targeting program managers were less influential compared to recommendations targeting high-level policy changes. Furthermore, we found that progress with the action plans was not uniform across facilities and counties. In the higher-level facilities (e.g. the County Referral Hospitals), progress was more pronounced compared to lower-level facilities such as the sub-county facilities and the health

centres. This can be explained by higher-level facilities having bigger budgets and receiving more funds from the Linda Mama scheme as they provide more maternal health services. In addition, the lack of infrastructure to support the processing of Linda Mama claims (such as lack of IT equipment, internet, etc.) put lower-level facilities at a disadvantage with the amount of reimbursement they can receive as they could not process claims for reimbursement quickly. Another persistent challenge was delayed reimbursement from NHIF for Linda Mama services. Delayed payments can affect the provision of services and promote out-of-pocket payments [37, 38]. These challenges can frustrate providers willing to bring about change but cannot do so, as the lack of resources limits their decision-making space. Such factors and their influence must be considered during the design of the iKT strategy.

A key challenge with our KT strategy was its sustainability. While the research team was highly involved initially, we could not continue this high-level engagement during the implementation refinement stage. In this study, the researchers' main roles were evidence generation through the process evaluation, dissemination of these findings and facilitating the co-development of action plans. To support implementation of the action plans, we collaborated with a partner (ThinkWell) to provide technical assistance and support to the implementers. However, the study participants expected continuous engagement with the researchers over the implementation duration, which was not sustainable due to resource constraints. Future studies should consider how best to plan for sustainability earlier in the KT process. Another sustainability issue was the high staff turnover at the county offices and facilities. This meant that out of the 105 participants who attended the workshops, only 25 were still at their stations and could implement the action plans and report on them. Facility-in-charges who were not at the workshop were unaware of the action plans. This highlights issues with the spread of the evaluation findings amongst the stakeholders, thus presenting a barrier to sustained change. Future iKT approaches for similar contexts must consider how new stakeholders are updated.

Few case studies exist that describe how iKT approaches can be implemented to support the use of program evaluation finds within health systems. This study contributes to growing body of literature by offering insights into how deliberate integration of KT in evaluation processes can enhance the practical use of evaluation findings in practice settings. This findings from this study can guide other researchers undertaking process evaluations by highlighting opportunities and possible pitfalls.

A limitation of the study is that it does not include quantitative data on the improvements made to Linda mama, which would have been helpful in the triangulation of data. However, as our main aim was to capture the perceptions of the usefulness and value of the iKT approach from the stakeholders, we felt that a qualitative approach would be sufficient. To build on our findings, future research can quantify stakeholder preferences for engaging in KT for improved uptake of evaluation findings. This would support researchers in designing responsive KT strategies to support evaluation use.

## Conclusion

Our findings report uptake of evaluation findings and suggest perceived improvements in the implementation of Linda Mama in 3 Kenyan counties through our iKT approach. However, these improvements were not uniform across and within the counties. Contextual factors such as the COVID-19 restrictions, lack of infrastructure and delayed reimbursement of funds hindered the full implementation of the evaluation findings. Overall, participants appreciated the value of the iKT approach and acknowledged its influence in supporting the use of the evaluation findings by promoting stakeholder engagement and providing a space for learning and sharing. This study contributes to the literature on utilising evaluation findings of complex,

multi-stakeholder health programs by demonstrating the value of iKT approaches in supporting use. When embedded throughout the evaluation, KT can significantly enhance the use and implementation of evaluation findings. Although we report positive outcomes with this approach, iKT strategies can be challenging. Future approaches should consider the impact of context and how to navigate it and how best to sustain change, especially given a high turnover of stakeholders in decision-making positions.

## Supporting information

**S1 Checklist. Inclusivity in global research.**
(DOCX)

**S1 Table. Progress and perceived outcomes of implementing action plans in the counties.**
(DOCX)

## Acknowledgments

We would like to acknowledge the support of Felix Murira, Shano Guyo, Nick Mwendwa, and Daniel Koech from ThinkWell Kenya; Fardosa Abdi from the National Hospital Insurance Fund; and the respective County Departments of Health for their administrative support throughout the study.

## Author Contributions

**Conceptualization:** Anooj Pattnaik, Nirmala Ravishankar, Edwine Barasa.

**Data curation:** Fatuma H. Guleid, Stacey Orangi, Angela Kairu, Brian Arwa, Anne Musuva.

**Formal analysis:** Fatuma H. Guleid, Stacey Orangi.

**Methodology:** Edwine Barasa.

**Project administration:** Ileana Vilcu.

**Supervision:** Edwine Barasa.

**Validation:** Stacey Orangi.

**Writing – original draft:** Fatuma H. Guleid.

**Writing – review & editing:** Stacey Orangi, Angela Kairu, Brian Arwa, Janet Keru, Anne Musuva, Ileana Vilcu, Anooj Pattnaik, Edwine Barasa.

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
