## [Decision Letter · Decision Letter 0]

29 Jan 2024

PGPH-D-23-02056

Using integrated knowledge translation to support the use of evaluation findings: A case study of the Linda Mama Free Maternity Program in Kenya

Dear Dr. Guleid,

Thank you for submitting your manuscript to PLOS Global Public Health. After careful consideration, we feel that it has merit but does not fully meet PLOS Global Public Health’s publication criteria as it currently stands. Therefore, we invite you to submit a revised version of the manuscript that addresses the points raised during the review process.

I concur with the assessment of both referees and believe that the manuscript can improve substantively from a round of revision. Please address the concerns raised by both referees and make necessary revisions.

We look forward to receiving your revised manuscript.

Kind regards,

Sarthak Gaurav

Academic Editor

Journal Requirements:

Additional Editor Comments (if provided):

Reviewers' comments:

Reviewer's Responses to Questions

**Comments to the Author**

1. Does this manuscript meet PLOS Global Public Health’s publication criteria? Is the manuscript technically sound, and do the data support the conclusions? The manuscript must describe methodologically and ethically rigorous research with conclusions that are appropriately drawn based on the data presented.

Reviewer #1: Yes

Reviewer #2: Yes

2. Has the statistical analysis been performed appropriately and rigorously?

Reviewer #1: N/A

Reviewer #2: Yes

3. Have the authors made all data underlying the findings in their manuscript fully available (please refer to the Data Availability Statement at the start of the manuscript PDF file)?

Reviewer #1: Yes

Reviewer #2: Yes

4. Is the manuscript presented in an intelligible fashion and written in standard English?

Reviewer #1: Yes

Reviewer #2: Yes

5. Review Comments to the Author

Reviewer #1: The paper claims that knowledge translation approaches are useful in improving the uptake of evaluation in the implementation of complex health programs with multiple stakeholders. the study is useful to the discipline moving beyond research findings more commonly found in literature to evaluation co-production.

Are the claims properly placed in the context of the previous literature? Have the authors treated the literature fairly?

The paper is placed within the growing field of the use of knowledge translation to increase uptake of evaluation findings. The authors do well to demonstrate the relevance of knowledge translation in evaluation work. The authors can improve by providing examples from the existing studies and describing their results and methodology briefly.

The authors demonstrate the usefulness of using knowledge translation for evaluation result uptake by describing the methodology thoroughly. the IKT approach is assessed based on actionable items, their progress as well the perceptions of implementors. The acknowledgment of the impact of Covid-19 and other contextual factors adds to the depth around the usefulness of IKT.

The methodology is detailed and allows for reproducibility. The manuscript is well organized and clea enough for non-expert to understand.

Reviewer #2: Manuscript review

The work is insightful for publication and contributes to the body of knowledge in maternal and child health

They have well written the Background describing the context and purpose of the study.

Methods

The use of the qualitative approach method (semi-structured questionnaire) makes it difficult for one to conclude how the saturation phenomenon was achieved in these questionnaires and how we can say that the opinions being expressed represent fully the reality (link with the selection criteria that should be done within a strategy to cover all aspects that are presented in the objectives list but also avoid biases)

It’s not clear how saturation was achieved in these semi-structured questionnaires given that half of the research participants dropped off during the study period and how we can say that the opinions being expressed represent fully the reality on the ground.

This is a qualitative study using a structured questionnaire. Why was only one method of data collection? Was it seen as exhaustive enough to capture the views of the participants? There are experiences that are well expressed in group interviews, which bring out different dimensions and group synergy, making the information rich. Not everyone is good at expressing their views in writing, but they express themselves well in oral narrations.

The study subjects

At the grassroots level, we have the community health workers, who play a critical role in sensitisation and health education at the community level and are the critical link between the facility and the mothers (consumers of the service). why were they not considered part of the study? On the other hand, we have mothers who are consumers of free maternity services. I am grappling on why they were not part of the study.

The study states the three counties as explained in the study area section, but in the findings they have a new label; for example, Isiolo is a study site but given a label as County A, making it confusing. I is important to maintain uniformity and sticking to the original label to give the reader a simple time

Table 1 is too long. It’s advisable if they subdivide the table based on themes guided by the study objectives instead of counties; this will allow easy comparisons of issues across the three study sites. For example, if administrative issues cut across the three countries, this can make a difference.

The discussion section needs to be strengthened by clearly stating the a) strengths and weaknesses of the study; b) strengths and weaknesses in relation to other studies that have been carried out c) discussion of important differences in results, meaning of the study, unanswered questions, and future research.

Conclusion: The conclusion should provide a brief summary of the key findings, potential implications and the way forward.

It’s not clear from the abstract what qualitative techniques were used to collect and analyse the data for this study

6. PLOS authors have the option to publish the peer review history of their article (what does this mean?). If published, this will include your full peer review and any attached files.

**Do you want your identity to be public for this peer review?** For information about this choice, including consent withdrawal, please see our Privacy Policy.

Reviewer #1: No

Reviewer #2: **Yes: **Beverly M. Ochieng

---

## [Decision Letter · Decision Letter 1]

9 Jul 2024

PGPH-D-23-02056R1

Using integrated knowledge translation to support the use of evaluation findings: A case study of the Linda Mama Free Maternity Program in Kenya

Dear Dr. Guleid,

Thank you for submitting your manuscript to PLOS Global Public Health. After careful consideration, we feel that it has merit but does not fully meet PLOS Global Public Health’s publication criteria as it currently stands. Therefore, we invite you to submit a revised version of the manuscript that addresses the points raised during the review process.

We look forward to receiving your revised manuscript.

Kind regards,

Anteneh Asefa Mekonnen, Ph.D.

Academic Editor

Journal Requirements:

1. “Please include a complete copy of PLOS’ questionnaire on inclusivity in global research in your revised manuscript. Our policy for research in this area aims to improve transparency in the reporting of research performed outside of researchers’ own country or community. The policy applies to researchers who have travelled to a different country to conduct research, research with Indigenous populations or their lands, and research on cultural artefacts. The questionnaire can also be requested at the journal’s discretion for any other submissions, even if these conditions are not met.  Please find more information on the policy and a link to download a blank copy of the questionnaire here: https://journals.plos.org/globalpublichealth/s/best-practices-in-research-reporting. Please upload a completed version of your questionnaire as Supporting Information when you resubmit your manuscript.”

Additional Editor Comments (if provided):

Reviewers' comments:

Reviewer's Responses to Questions

**Comments to the Author**

1. If the authors have adequately addressed your comments raised in a previous round of review and you feel that this manuscript is now acceptable for publication, you may indicate that here to bypass the “Comments to the Author” section, enter your conflict of interest statement in the “Confidential to Editor” section, and submit your "Accept" recommendation.

Reviewer #2: All comments have been addressed

Reviewer #3: (No Response)

Reviewer #4: All comments have been addressed

2. Does this manuscript meet PLOS Global Public Health’s publication criteria? Is the manuscript technically sound, and do the data support the conclusions? The manuscript must describe methodologically and ethically rigorous research with conclusions that are appropriately drawn based on the data presented.

Reviewer #2: Yes

Reviewer #3: Partly

Reviewer #4: Yes

3. Has the statistical analysis been performed appropriately and rigorously?

Reviewer #2: Yes

Reviewer #3: N/A

Reviewer #4: I don't know

4. Have the authors made all data underlying the findings in their manuscript fully available (please refer to the Data Availability Statement at the start of the manuscript PDF file)?

Reviewer #2: Yes

Reviewer #3: Yes

Reviewer #4: No

5. Is the manuscript presented in an intelligible fashion and written in standard English?

Reviewer #2: Yes

Reviewer #3: Yes

Reviewer #4: Yes

6. Review Comments to the Author

Reviewer #2: The manuscript has been revised, all corrections incorporated

Reviewer #3: Thank you for the opportunity to review this paper which presents an interesting contribution to the potential use of a knowledge translation/integrated knowledge translation approach in evaluation studies within healthcare. Generally, I found this to be an interesting study but it is perhaps trying to demonstrate too many areas in one paper. My sense is that it is doing four things: 1. Report on the design of an evaluation; 2. Report on the outcome of an evaluation; 3. Show how a partnership approach to evaluation was utilised; 4. Show how an IKT approach formed the basis of an evaluation. The introduction and the conclusion are reasonably focused on the use of KT/IKT as an approach but otherwise the paper veered between all of these four things and potentially lost some focus as a result.

There are some specific areas where I feel that the paper could be more focused and where clarifications are needed. They are as follows:

- It would be useful to define KT when it is first introduced as a concept as there are myriad definitions and terms in existence. It appears the authors are using Strauss et al as their preferred reference point but that is not clear. Also, on page 3 they have referred to the purpose of KT as promoting research uptake. However, the purpose is somewhat broader than they have suggested i.e. to improve health, the provision of effective services, to enhance the use of research in practice, policy, guidance etc. and to potentially stop doing what is ineffective and wasteful. In essence, it bridges the gap between research knowledge and its application in practice.

- It is not clear if the research team used KT/IKT as a tool within the evaluation framework or simply saw it as an approach. An obvious limitation of the study would be a failure/lack of integration of the KT approach into the evaluation design. The approach as described seems to be very much focused on KT in mobilising findings when in fact KT starts much earlier in the process.

- A minor point of semantics but on page 4 paragraph 3 refers to a ‘free maternity policy’, I presume they mean a ‘policy providing free maternity health care’.

- On page 5 the heading ‘co-research’ did not seem to reflect the activity usually associated with the term. Co-research usually suggests that co-researchers have been part of the research team throughout the project but in this study their role seems to have been linked to two key activities pre and post implementation of the Linda Mama programme. Step 1 states that stakeholders were asked to identify key research questions. Presumably this activity may have generated a number of questions but it is not clear if they were all used or whether there was a process for selecting those that were deemed most relevant or important.

- Were the participants in the study and the stakeholders references throughout the paper the same people or are they different groups? It is not clear if there is a distinction or whether the study participants were actively engaged with the stakeholders who were implementing the evaluation findings. Similarly, on page 85, it was unclear who the stakeholders were and whether they included some of those identified in the previous section as being barriers to implementation.

- The results section did not seem to be focused sufficiently on which aspects of IKT worked. A series of quotes was included which, whilst interesting, did not drill down into which of the technical aspects of IKT worked. There was also very little in relation to the impact and sustainability aspects of KT. It was not clear if the IKT elements of the evaluation activity actually provided any training or information about IKT for the participants, how it added value to their implementation process and how to use IKT to aid more effective information sharing and implementation. The use of one or more KT theories, models or frameworks may have been helpful to the researchers in this context but there is no indication that this was a consideration for them.

- In the discussion the authors mention the participants wish to have the researchers engaged to a greater degree to foster sustainability but the researchers role was limited to evidence generation and they could not maintain continuous engagement. This should be cited as a limitation of the study as a key component of KT is to establish who is engaged in the KT process (i.e. stakeholders), how and when and how the activity will be sustained. The research teams lack of capacity and not planning for who would maintain the KT activity is something for the authors to reflect on in terms of using (or partially adopting) a KT approach.

There are some additional references and sources of information the authors may find useful:

The CIHR guide to KT planning and IKT.

Nguyen et al 2020.

Kothari and Wathen 2017.

Lawrence et al 2019.

I hope these comments are helpful to the authors in revising their paper.

Reviewer #4: Overall, the authors addressed the previous peer reviewer comments for the most part. That said, I have comments based on how the revisions were handled.

• Data – plural (throughout manuscript)

• Abstract describes: case study approach using qualitative approach. In another section “participatory approach” (used twice) and “KT approach”. Can the word “approach” be limited?

• I found the word approach confusing because it was used to describe different things in the same paragraph and sentence overall. The revisions added in the word “approaches”

• Page 3 – “while the literature in both fields has grown”

• Some grammar issues with added text (page 3) –

• Page 26 – Discussion – where the revision was made – it has been reported to increase what type of use?

• Page 27 – Revisions need to be copyedited for grammatical issues

• Re: addressing limitations –might the the depth of information from the interviews provided preliminary evidence that might be explored in different ways (using quantitative data, and/or further qualitative data gathering) in the future?

• While the comments have been addressed – identifying the benefit of this work to add to the literature, and support integration between KT and evaluation use would be beneficial. It might be possible to add references to this at the beginning of the discussion – this emphasizes why the sample was chosen for the interviews

• How were the qualitative data analyzed. This is still not completely clear. What strategy was used for this? When I read the description of the analysis I am not totally sure whether the data were coded based on the interview guide? (e.g., the 3 key issues of investigation?)? You also refer to a thematic framework that was developed by the FHG and SO (is FHG – a person?)

The qualitative data findings do not seem to reveal the thematic framework. Are there themes and do they relate to the 3 key issues of investigation?

• I agree with reviewer 2 that the Table 2 is extremely lengthy and hard to follow. Is it possible to add Table 2 to an addendum, but shorten table 2 for the manuscript to include the essential elements? (i.e., make it shorter and easier for a reader to take in on less space)?

Overall, comments were addressed to some extent, but still feel may need some additional revision.

7. PLOS authors have the option to publish the peer review history of their article (what does this mean?). If published, this will include your full peer review and any attached files.

**Do you want your identity to be public for this peer review?** For information about this choice, including consent withdrawal, please see our Privacy Policy.

Reviewer #2: **Yes: **Beverly Ochieng

Reviewer #3: **Yes: **Dr Virginia Minogue

Reviewer #4: No

---

## [Decision Letter · Decision Letter 2]

1 Oct 2024

PGPH-D-23-02056R2

Using knowledge translation to support the use of evaluation findings: A case study of the Linda Mama Free Maternity Program in Kenya

Dear Dr. Guleid,

Thank you for submitting your manuscript to PLOS Global Public Health. After careful consideration, we feel that it has merit but does not fully meet PLOS Global Public Health’s publication criteria as it currently stands. Therefore, we invite you to submit a revised version of the manuscript that addresses the points raised during the review process.

As the reviewer suggested, please submit a revised version after careful and thorough proofreading. 

We look forward to receiving your revised manuscript.

Kind regards,

Anteneh Asefa Mekonnen, Ph.D.

Academic Editor

Journal Requirements:

Additional Editor Comments (if provided):

1. Please include the following request in the decision letter, and ping me with follow-up. “Please include a complete copy of PLOS’ questionnaire on inclusivity in global research in your revised manuscript. Our policy for research in this area aims to improve transparency in the reporting of research performed outside of researchers’ own country or community. The policy applies to researchers who have travelled to a different country to conduct research, research with Indigenous populations or their lands, and research on cultural artefacts. The questionnaire can also be requested at the journal’s discretion for any other submissions, even if these conditions are not met.  Please find more information on the policy and a link to download a blank copy of the questionnaire here: https://journals.plos.org/globalpublichealth/s/best-practices-in-research-reporting. Please upload a completed version of your questionnaire as Supporting Information when you resubmit your manuscript.

Reviewers' comments:

Reviewer's Responses to Questions

**Comments to the Author**

1. If the authors have adequately addressed your comments raised in a previous round of review and you feel that this manuscript is now acceptable for publication, you may indicate that here to bypass the “Comments to the Author” section, enter your conflict of interest statement in the “Confidential to Editor” section, and submit your "Accept" recommendation.

Reviewer #3: All comments have been addressed

2. Does this manuscript meet PLOS Global Public Health’s publication criteria? Is the manuscript technically sound, and do the data support the conclusions? The manuscript must describe methodologically and ethically rigorous research with conclusions that are appropriately drawn based on the data presented.

Reviewer #3: Yes

3. Has the statistical analysis been performed appropriately and rigorously?

Reviewer #3: N/A

4. Have the authors made all data underlying the findings in their manuscript fully available (please refer to the Data Availability Statement at the start of the manuscript PDF file)?

Reviewer #3: Yes

5. Is the manuscript presented in an intelligible fashion and written in standard English?

Reviewer #3: Yes

6. Review Comments to the Author

Reviewer #3: I am satisfied that the authors have addressed my comments. There are some minor issues with grammer and spelling that can be rectified with further proof reading.

7. PLOS authors have the option to publish the peer review history of their article (what does this mean?). If published, this will include your full peer review and any attached files.

**Do you want your identity to be public for this peer review?** For information about this choice, including consent withdrawal, please see our Privacy Policy.

Reviewer #3: **Yes: **Dr Virginia Minogue

---

## [Editor Report · Decision Letter 3]

30 Oct 2024

Using knowledge translation to support the use of evaluation findings: A case study of the Linda Mama Free Maternity Program in Kenya

PGPH-D-23-02056R3

Dear Ms Guleid,

We are pleased to inform you that your manuscript 'Using knowledge translation to support the use of evaluation findings: A case study of the Linda Mama Free Maternity Program in Kenya' has been provisionally accepted for publication in PLOS Global Public Health.

Best regards,

Anteneh Asefa Mekonnen, Ph.D.

Academic Editor